Genome-wide identification and expression analysis of the EXO70 gene family in grape (Vitis vinifera L)

Wang Han
Ma Zong-Huan
Mao Juan
Chen Bai-Hong bhch@gsau.edu.cn
Department of Horticulture, Gansu Agricultural University , Lanzhou , China
Okpala Charles
Electronic publication date: 2021 Apr 21
Publication date: 2021
Volume: 9
Electronic Location ID: e11176
Received 2020 Sep 11; Accepted 2021 Mar 8
Copyright: ©2021 Wang et al.
Copyright year: 2021
Copyright holder: Wang et al.
License: This is an open access article distributed under the terms of the Creative Commons Attribution License, which permits unrestricted use, distribution, reproduction and adaptation in any medium and for any purpose provided that it is properly attributed. For attribution, the original author(s), title, publication source (PeerJ) and either DOI or URL of the article must be cited.
License URL: https://creativecommons.org/licenses/by/4.0/

Keywords: EXO70 gene, Grape, Gene expression, qRT-PCR

Funding: Public Recruitment of Doctoral Research Funding Gansu Agricultural University , China GAU-KYQD-2019-18 Discipline Construction Funds for Horticulture, Gansu Agricultural University, China GSAU-XKJS-2018-226 This research was funded by Public Recruitment of Doctoral Research Funding Gansu Agricultural University, China (GAU-KYQD-2019-18) and Discipline Construction Funds for Horticulture, Gansu Agricultural University, China (GSAU-XKJS-2018-226). The funders had no role in study design, data collection and analysis, decision to publish, or preparation of the manuscript.

==============================
EXO70 is the pivotal protein subunit of exocyst, which has a very crucial role in enhancing the shielding effect of the cell wall, resisting abiotic and hormonal stresses. This experiment aims to identify family members of the EXO70 gene family in grape and predict the characteristics of this gene family, so as to lay the foundation of further exploring the mechanism of resisting abiotic and hormone stresses of VvEXO70s. Therefore, the Vitis vinifera ‘Red Globe’ tube plantlet were used as materials. Bioinformatics was used to inquire VvEXO70 genes family members, gene structure, system evolution, cis-acting elements, subcellular and chromosomal localization, collinearity, selective pressure, codon bias and tissue expression. All of VvEXO70s had the conserved pfam03081 domain which maybe necessary for interacting with other proteins. Microarray analysis suggested that most genes expressed to varying degrees in tendrils, leaves, seeds, buds, roots and stems. Quantitative Real-Time PCR (qRT-PCR) showed that the expression levels of all genes with 5 mM salicylic acid (SA), 0.1 mM methy jasmonate (MeJA), 20% PEG6000 and 4 °C for 24 h were higher than for 12 h. With 20% PEG6000 treatment about 24 h, the relative expression of VvEXO70-02 was significantly up-regulated and 361 times higher than CK. All genes’ relative expression was higher at 12 h than that at 24 h after treatment with 7 mM hydrogen peroxide (H2O2) and 0.1 mM ethylene (ETH). In conclusion, the expression levels of 14 VvEXO70 genes are distinguishing under these treatments, which play an important role in the regulation of anti-stress signals in grape. All of these test results provide a reference for the future research on the potential function analysis and plant breeding of VvEXO70 genes.

Introduction

The yield and quality of plants under biotic and abiotic stresses are seriously damaged, so it is essential to explore the defense mechanism (Bu et al., 2019). When plants are under biotic and abiotic stresses, the signaling pathway related to receptor proteins is activated and exocytosis occurs, which enhances the shielding effect of cell wall (Yun & Kwon, 2017). Exocytosis refers to the process of the transport vesicles derived from the post-golgi transport membrane fuse with the target membrane pass through certain transport pathways (Jurgens & Geldner, 2002). It plays a very important role in all eukaryotic cells, including cell growth, cell polarization, cell division, cell information transfer, the formation of the cell walls of the physiological process (Yao et al., 2013; Yang et al., 2013; Yang et al., 2015a; Yang et al., 2015b), especially has a crucial role in plant resistance mechanisms (Gu, Zavaliev & Dong, 2017).

EXO70 is one of the eight protein subunits in the exocrine complex, with the seven other ones (SEC3, SEC5, SEC6, SEC8, SEC10, SEC15 and EXO84), is involved in the tethering process of the membrane vesicles at the budding site. The tethering process is the third step (budding, transport, tethering, fusion) in the material exchange process of different membrane structures in the cell, which is involved in the initial contact process of the transport vesicles and the target membrane and is a key step in regulating extracellular secretion (TerBush & Novick, 1995; TerBush et al., 1996; Finger, Hughes & Novick, 1998; Guo, Grant & Novick, 1999; Whyte & Munro, 2002; Elias et al., 2003; Cai, Reinisch & Ferro-Novick, 2007). The C-terminal of EXO70 protein carries several negative charge residues (Zhu, Wu & Guo, 2019), and EXO70 and SEC3 can bind to phosphatidylinositol-4, 5-diphosphate (PIP2), which is involved in the anchoring of exocrine complex secretory tomont and target plasma membrane (Liu et al., 2018). Meanwhile, EXO70 can directly interact with Rho GTPase family members of small G protein, participating in the assembly and activation of SNARE protein and regulating the assembly of exocyst complex (Sivaram et al., 2005; He et al., 2007; Moore, Robinson & Xu, 2007; Wu et al., 2010; Chen et al., 2012; Ren & Guo, 2012).

The mechanism of exocyst was first identified in yeast studies. EXO70 was localized at the activation site of vesicle-plasma membrane fusion in yeast (Mathieson et al., 2010). In animals, eight proteins of yeast-homologous exocrine complex were first isolated in the brain tissues of mice (Ma et al., 2016). EXO70 gene families of yeast and animals have only one member, meanwhile, there are multiple members in EXO70 gene family of plants, which shows the multi-copy phenomenon peculiar to plants (Table S1) (Moore, Robinson & Xu, 2007; Yang et al., 2013). For example, the genes encoding SEC6, SEC8 and SEC10 in the Arabidopsis thaliana genome have only one copy respectively, while the genes encoding EXO70 have 23 copies. The genes encoding SEC5, SEC6, SEC8 and SEC10 in the rice genome had only one copy each, while the genes encoding EXO70 had 47. This indicates that the replication of EXO70 gene is unique to terrestrial higher plants (Li et al., 2010). The cause of this phenomenon has not been determined, and it is speculated that the EXO70 genes in different organisms are involved in various biological processes (Bu et al., 2019). Studies have found that exocytosis of higher plants acts on plant growth and development (Jurgens & Geldner, 2002). Among the 23 A. thaliana EXO70 genes, AtEXO70A1 was involved in the growth of pollen tube and root hairs, resulting in shorter root and stigma hairs (Synek et al., 2006; Samuel et al., 2009). AtEXO70C1 (At5g13150) gene mutated, resulting in delaying pollen tube development and blocking male transmission (Li et al., 2010). Moreover, EXO70 is involved in pollen-stigma interactions in Brassica and the over-expression of BrEXO70A1 is sufficient to overcome the self-pollination rejection of the species with partial self-incompatibility (Samuel et al., 2009; Zhang et al., 2010). During the development and maturation of Glycine max L, the expression of some GmExo70J genes including GmExo70J1, GmExo70J6 and GmExo70J7 increases greatly in floral organ-supporting receptacles, indicating a possible role in seed development (Wang et al., 2016).

At present, the EXO70 gene in yeast and mammals has been widely studied, but only a few kinds of plants have been done, such as tomato (Solanum lycopersicum) had 22, potato (Solanum tuberosum) had 21, tobacco (Nicotiana benthamiana) had 44 (Yu et al., 2018), A. thaliana had 23 (Li et al., 2010), Chinese cabbage (Brassica pekinensis) had 39, cabbage (Brassica oleracea) had 45 (Yang et al., 2015b), diploid tobacco had 24 (Juraj et al., 2017), wheat (Triticum aestivum) had 200 (Zhao et al., 2018) EXO70 genes. However, the study of EXO70 gene families in grape is rarely reported. Besides, genome-wide identification and expression analysis are effective ways to clarify the classification and composition of gene family members in genome, which is the primary task to explore biological issues related to species characteristics. These can lay a foundation for subsequent functional studies and genetic manipulation of gene. Therefore, in this study, a genome-wide identification and expression analysis of VvEXO70 gene family was conducted and bioinformatics was used to inquire VvEXO70 gene family members, genetic structure, system evolution and cis-acting element, subcellular and chromosomal localization, collinearity, selective pressure, codon bias and tissue expression. By establishing the simulation environment of different treatment for grape test-tube seedlings and analyzing the relative expression quantity, it could provide the reference for the further research of the function of the gene family and the breeding of resistant varieties in grape.

Materials and Methods Introduction

Overview of study

As shown in Fig. 1, the experiment mainly included genome-wide identification and expression analysis of EXO70 gene family in grape to preliminarily predict the function of VvEXO70s. On the one hand, we identified the members of VvEXO70s by bioinformatics and on the other hand, we analyzed the responses of this gene family to some abiotic stresses by qRT-PCR. These test results could provide a reference for the future research on the potential function analysis and plant breeding of VvEXO70 genes.

Figure 1 The research route of this study.

The experiment mainly included genome-wide identification and expression analysis of EXO70 gene family in grape to preliminarily predict the function of VvEXO70s.

Plant materials and treatments

The experiment materials were Vitis vinifera ‘Red Globe’ tube seedings, which were stored in College of Horticulture, Gansu Agricultural University. At the beginning of the test, we placed stem-segment with single bud of tube seedlings on ordinary GS medium and pH was 5.8∼6.0. Materials were cultured in the incubator under a 16 h of light/8 h of dark cycle at 25 °C/23 °C. Different stress treatments were conducted about 35 d seedling plants. For abiotic stress treatments, the seedlings were incubated in a solution containing 20% PEG6000, 10 mM hydrogen peroxide (H2O2) and the low temperature treatment condition is 4 °C. For phytohormone treatments, the seedlings were cultured in GS medium with 5 mM salicylic acid (SA), 0.1 mM methy jasmonate (MeJA), 0.1 mM ethylene (ETH) and 0.2 mM abscisic acid (ABA) respectively, meanwhile, the volume of distilled water as a CK. Every stress repeated three times. Then we collected leaves from each treatment about 12 h and 24 h. All of the collected materials were stored at −80 °C for RNA extraction and gene expression analysis.

Identification of EXO70 genes family members in grape and analysis of physicochemical properties

A. thaliana IDs (Li et al., 2010; Zhao et al., 2018) were used to download amino acid sequences of each gene from TAIR database (https://www.arabidopsis.org). After that amino acid sequences were used to homologously search EXO70 genes family in grape genome website (http://www.genoscope.cns.fr/) and rice genome website (http://rice.plantbiology.msu.edu/). We could screen genes by judging whether they include the functional domain ‘PF03081.15’ on HMMER (https://www.ebi.ac.uk/Tools/hmmer/). Meanwhile, DNAMAN software could do multiple sequence alignment. ExPASy (https://web.expasy.org/protparam/) was used to acquire some basic physicochemical properties about VvEXO70s such as the number of exon, molecular weight (MW), isoelectric point (pI), instability index (I.I), aliphatic index (A.I), grand average of hydropathicity (GRAVY).

Bioinformatics analysis of the VvEXO70s

Clustal X, MEGA7.0 and EvolView (https://www.evolgenius.info/) were used to constructed a phylogenetic tree of grape, rice and A. thaliana EXO70 genes by the basis of the neighbor-joining method. The bootstrap analysis was performed using 1,000 repetitions. The gene structure of the VvEXO70s was detected by using GSDS 2.0 (http://gsds.cbi.pku.edu.cn/) based on their corresponding amino acid sequences. The secondary structure was identified on NPS@:SOPMA (https://npsa-prabi.ibcp.fr/cgi-bin/npsa_automat.pl?page=npsa_sopma.html). The 3D structure was created with SWISS-MODEL (https://swissmodel.expasy.org/). The chromosomal location information of VvEXO70s was obtained from the grape genome website (http://www.genoscope.cns.fr/) and was drawn a picture by MG2C (http://mg2c.iask.in/mg2c_v2.0/). MEME (http://meme-suite.org/tools/meme) was used to analyze the conserved motifs of VvEXO70 proteins. Subcellular localization analysis was conducted on WoLF PSORT (https://www.genscript.com/wolf-psort.html). PLACE (https://www.dna.affrc.go.jp/PLACE/?action=newplace) was used to analyze the distribution of cis-acting elements of the 2 kb of upstream of 14 VvEXO70 genes.

Collinearity and selective pressure analysis VvEXO70s

Online software (http://tools.bat.infspire.org/circoletto/) was used to analyze the collinearity of EXO70 genes between grape and A. thaliana.

Pairs of genes with collinearity would be done sequence alignment by using Clustal Omega (https://www.ebi.ac.uk/Tools/msa/clustalo/). PAL2NAL (http://www.bork.embl.de/pal2nal/index.cgi?) was used to calculate the non-synonymous/synonymous (dN/dS) value of duplicate gene pairs (Goldman & Yang, 1994).

Codon usage bias analysis VvEXO70s

Codon composition and usage preference of VvEXO70 gene family members were analyzed by CodonW software. Relative synonymous codon usage (RSCU) refers to the ratio of the frequency used by a particular codon to the frequency expected in unbiased use. Measures of codon use preference include: RSCU, number of valid codons (ENC), codon adaptation index (CAI), and codon bias index (CBI). Measures indicators of codon composition include: the occurrence frequency of adenine, thymine, guanine, and cytosine at position 3 (A3s, T3s, G3s, C3s), frequency of optimal codons (FOP), Guanine and cytosine content (GC content), GC content at the third site of the synonymous codon (GC3s content), aromatic amino acid frequency (Aromo), synonymous codon number (L_sym), the total number of synonymous and non-synonymous codons (L_aa), protein hydrophilicity (Gravy). Correlation analysis between codon composition and preference parameters was carried out by using SPSS 23.0 statistical software.

Tissue-specific expression patterns of VvEXO70s

BLAST program on Ensembl Plants (http://plants.ensembl.org/index.html) was used to find the accession numbers of tissue-specific expression of VvEXO70s. These ids were used to gain the tissue-specific expression data related to the different tissues at various developmental stages of grape on BAR (https://bar.utoronto.ca/). R language was used to visualize the results.

RNA Isolation and qRT-PCR analysis of VvEXO70s

The RNA were extracted from plant leaves by using a Spectrum Plant Total RNA kit (Sigma St. Louis, MO, USA) following the operating instructions.

The primer design and synthetic were accomplished in Sangon Biotech in Shanghai. The Reverse Transcriptase M-MLV (RNase H-) kit (TaKaRa Biotechnology. Lanzhou, China) was used to synthesize cDNA. Light Cycler® 96 Real-Time PCR System (Roche, Basel, Switzerland) was used to perform qRT-PCR of VvEXO70s. The gene primers (Table S2) were designed in Sangon and used for PCR amplification, among which GAPDH gene (GenBank accession no. CB973647) was internal control genes. The amplification volume was 25 µL containing 1 µL forward primer, 1 µL reverse primer, 1.5 µL cDNA, 9 µL ddH2O and 12.5 µL TaKaRa SYBR Premix Ex Taq. II (TaKaRa Biotechnology. Lanzhou, China). The response procedures was: 95 °C for 30 s, 40 cycles of 95 °C for 5 s, and 60 °C for 30 s. For melting curve analysis, a program including 95 °C for 15 s, followed by a constant increase from 60 °C to 95 °C, was included following the PCR cycles. Each treatment was run three biological replicates. The 2−ΔΔCT method was used to calculate the relative expression levels of genes (Udvardi, Czechowski & Scheible, 2008).

Figure 2 Distribution of VvEXO70 gene family on chromosome.

Chromosomal locations of 14 VvEXO70 genes. The red stripe represents some of the chromosomes in the grape, and the chromosome number is showed at the top of each red stripe.

Results

Identification and physicochemical characterization of VvEXO70s

Fourteen VvEXO70 genes were homologously searched and named as VvEXO70-01 ∼VvEXO70-14. Chromosome localization (Fig. 2) showed that the location of VvEXO70-02 was unknown and the other 13 genes located on eight chromosomes of grape, named Chr01st, Chr04th, Chr06th, Chr08th, Chr09th, Chr13th, Chr14th and Chr17th. Both VvEXO70-09 and VvEXO70-10 were located on the same site of Chr09th and belonged a pair of tandem duplication genes. The physicochemical properties of VvEXO70s suggested that the number of amino acids mainly were concentrated in 610∼690 aa, but among them VvEXO70-13 had 452 aa. In all genes, the number of MW of VvEXO70-06 (77426.01 kD) was the largest and the minimum is VvEXO70-07 (51585.27 kD). Also, the number of exons was mainly 1, 2, and 3, but the exons of VvEXO70-02 and VvEXO70-05 were 12 and 11, respectively (Table 1).

Analysis of gene structure and subcellular localization of VvEXO70s

Secondary structural analysis (Table S3) showed that the VvEXO70 gene family was mainly composed of alpha-helix and random coil, and the proportion of the two was similar, with the proportion of beta-turn, not more than 10%. The tertiary structure showed that VvEXO70-09, VvEXO70-10 and VvEXO70-12 were similar and greatly different from the rest of the genes (Fig. S1).

Subcellular localization (Table S4) found that most of the EXO70 proteins may be locate in the nucleus, cytoplasm, chloroplasts, mitochondria, and plasma membrane. Except for VvEXO70-08 and VvEXO70-11, other proteins would locate in the cytoplasm and nucleus, among which VvEXO70-05 was possibly the most abundant in the cytoplasm, and VvEXO70-01 and VvEXO70-07 were potentially the richest in the nucleus. VvEXO70-08 scored the highest probability to be present only in the chloroplast, In the mitochondria, VvEXO70s maybe have low content. VvEXO70-01 and VvEXO70-03 only potentially existed in peroxidases. VvEXO70-03 and VvEXO70-04 only possibly located in cytoskeleton. There were six genes would locate in the plasma membrane, among which VvEXO70-11 was possibly the most abundant. VvEXO70-06 and VvEXO70-13 could locate in the extracellular matrix. There was only one gene possibly located in the nuclear and nuclear and plasma membrane respectively, and these were VvEXO70-04 and VvEXO70-05. We predicted that genes locating in different organelles have different physiological functions, such as photosynthesis, growth and development and the like.

Table 1 Physicochemical property analysis of VvEXO70 gene family.

Gene	Gene Accession No.	Length of CDS	Length of genomic	Length of amino acid	Exon	Molecular weight	Isoelectric point	Instability index	Aliphatic index	Grand average of hydropathicity	
VvEXO70-01	GSVIVP00006977001	1920	1965	639	2	72341.22	5.18	53.78	72.18	−0.554	
VvEXO70-02	GSVIVP00013809001	1953	33699	650	12	73686.77	8.12	45.36	84.03	−0.443	
VvEXO70-03	GSVIVP00016199001	1932	1932	643	1	73525.06	5.56	46.58	96.30	−0.304	
VvEXO70-04	GSVIVP00017936001	1974	1974	657	1	74221.00	4.89	47.88	87.88	−0.358	
VvEXO70-05	GSVIVP00021515001	1947	8150	648	11	73278.55	7.34	44.58	86.59	−0.406	
VvEXO70-06	GSVIVP00023109001	2067	2085	688	2	77426.01	6.78	57.26	78.43	−0.344	
VvEXO70-07	GSVIVT00024003001	1363	1733	452	1	51585.27	6.74	49.77	94.49	−0.224	
VvEXO70-08	GSVIVP00024408001	2067	2085	621	1	70148.30	6.21	53.02	78.68	−0.362	
VvEXO70-09	GSVIVP00025283001	2052	7574	683	2	77236.89	7.84	41.42	98.78	−0.178	
VvEXO70-10	GSVIVP00025287001	1971	2007	656	2	74589.93	6.60	40.11	100.93	−0.170	
VvEXO70-11	GSVIVP00029521001	2025	8680	674	3	76259.83	6.62	44.74	94.33	−0.023	
VvEXO70-12	GSVIVP00032320001	1899	2004	632	2	71019.17	6.85	42.21	106.65	−0.108	
VvEXO70-13	GSVIVP00034051001	1836	1836	611	1	69560.89	5.34	50.30	92.13	−0.301	
VvEXO70-14	GSVIVP00038035001	1959	2491	652	1	73953.32	4.82	51.84	90.90	−0.370	
	

Phylogenetic evolution, Multiple sequence alignment and motif analysis of VvEXO70s

In this study, all 77 EXO70 protein sequences of grape (14), A. thaliana (23) and rice (40) were used to construct a phylogenetic tree. Figure 3 showed that they were divided into 10 sub-groups, namely EXO70A ∼J, which consisted 11, 6, 5, 7, 9, 11, 16, 2, 8 and 2 members respectively. Sub-group EXO70A, B, C, D, E, and G contained grape, A. thaliana and rice genes. EXO70F, H and J did not contain the VvEXO70 genes but only rice genes. Meanwhile, Fig. 4A showed that the groups of the separate cluster analysis of VvEXO70 only had seven ones.

Figure 3 Grape, Arabidopsis thaliana, rice evolution tree of EXO70 gene family.

Grape, A. thaliana, rice evolution tree of EXO70 gene family. White circle, triangle, star represent grapes, A. thaliana, and rice EXO70 genes, respectively. Lines and background with different colors represent different subgroups.

Figure 4 Gene structure, motif and multiple sequence analysis of VvEXO70 gene family.

(A) Cluster analysis of Vv EXO70 s. (B) Genes structure of Vv EXO70 s. The exon, intron and upstream/downstream are represented by red boxes, gray lines and black boxes. (C) Protein motif. Different color represented different motif. (D) Multiple sequence result.

Besides, Fig. 4B showed that all VvEXO70 genes had different structures. VvEXO70-03, VvEXO70-04, VvEXO70-08, and VvEXO70-13 had only CDS fragments, no upstream and downstream gene sequences and introns. VvEXO70-09 and VvEXO70-14 had the upstream gene sequence. VvEXO70-02 and VvEXO70-05 had the downstream gene sequence. VvEXO70-07 was the only one gene containing upstream and downstream sequences in all genes. VvEXO70-02 had the longest fragment length that was more than 33 kb. Moreover, cluster analysis of genes with similar structure was found in the same subgroup.

In this experiment, 8 motifs were constructed and named motif1 ∼8. The result (Fig. 4C) showed that VvEXO70-07 did not have motif8, VvEXO70-12 had no motif3. This was consistent with the result of multiple sequence alignment (Fig. 4D), which showed that VvEXO70-12 had no complete motif3 and the motif8 of VvEXO70-07 was incomplete. The remaining genes contained all motifs. It indicated that the gene family structure was similar and had certain conservatism in evolution.

Protein sequences of the VvEXO70 family were compared and some regions were intercepted (Fig. 4D). It mainly contained the specific structure domain ‘PF03081.15’ with base loci from about 250∼650 aa. Within its domain, it contained 15 highly conserved loci representing by red boxes in the Fig. 4D. At the same time, two highly fractured zones were found in the conservative structure domain, which were named by A and B in it, we speculated that the gene family had some variation during the evolutionary process.

cis-acting element analysis of VvEXO70s

The analysis of cis-acting elements is very important to understand the regulatory mechanism of genes. The online plant-care software tool was used to analyze the codon in the promoter sequences of 2 kb upstream of VvEXO70 gene, and it was found that the regulatory elements of the VvEXO70s were very abundant in number and variety. Main 24 types of hormone and stress related cis-acting regulatory elements were identified in the promoters of VvEXO70 genes (Fig. 5). However, we found nothing about the functional components of VvEXO70-02 on the online site. Emphasis was placed on the analysis of cis-acting elements in VvEXO70 genes related to grape growth and development, hormones and abiotic stress response. There are mainly the following types.

Figure 5 cis-acting elements existed in the 2 kb upstream region of VvEXO70 gene family.

The distribution of the 24 c is-acting elements in the promoter sequence of 14 VvEXO70 genes, shown in different colors. ABRE, cis-acting element involved in the abscisic acid responsiveness. ARE (anaerobic induction). AuxRR-core (auxin responsiveness). Box-4 (part of a conserved DNA module involved in light responsiveness). CAT-box (meristem expression). CGTCA-motif (MeJA-responsiveness). DRE (Drought). ERE (ETH). G-Box (light responsiveness). GARE-Motif (gibberellin-responsive). LTR (low-temperature responsiveness). MBS (MYB binding site involved in drought-inducibility). MYB (Drought). O2-site (zein metabolism regulation). P-box (gibberellin-responsive). STRE (Pressure). as-1 (SA). TCA-element (SA). TGA-element (auxin-responsiv). TGACG-Motif (MeJA-responsiveness). WUN-motif (callus). Circadian (circadian control).

cis-acting element involved in light responsiveness ; Box-4, G-box. cis-acting element involved in hormone: CGTCA-motif and TGACG-motif are related to MeJA responsive element. GARE-motif and P-box are related to Gibberellin-responsiveness responsive element. TCA-element and as-1 are related to SA responsive element, as well as Auxin responsive element and Abscisic acid responsive element. In addition, there are cis-acting regulatory element involved in circadian control, cell cycle regulation and abiotic stress, including low temperature responsive element, WUN-motif element and anaerobic induction element, MYB binding site involved in drough-induibility and so on. Figure 5 and Table S5 showed that all of VvEXO70s except VvEXO70-02 contained different species numbers of cis-acting elements. 13 VvEXO70 genes contained G-box and Box-4 elements associated with light stress and MYB and MYC elements associated with drought. In addition, there are 28, 30 and 40 elements related to SA, MeJA and ET. The results suggested that the VvEXO70s were involved in regulating various hormones and abiotic stresses to cope with various adverse environments.

Analysis of collinearity and selective pressure between grape and A. thaliana EXO70 genes

The online software was used to obtain the collinearity relationship of EXO70 genes between grape and A. thaliana (Fig. 6). It was found that there were 23 pairs of collinearity genes between grape and A. thaliana. Red indicated the largest similarity, followed by orange, green and blue, indicating the decreasing similarity. For example, in the red ribbon, VvEXO70-02 and AtEXO70A1, VvEXO70-05 and AtEXO70A2, VvEXO70-12 and AtEXO70G1, VvEXO70-13 and AtEXO70D2 had the strongest collinearity. There were 6 pairs of genes with collinear relationship on orange degree. The next weaker collinearity was found in 11 pairs of genes. However, the two weakest pairs were VvEXO70-08 and AtEXO70H6, VvEXO70-08 and AtEXO70H8. In addition, the collinear relationship between VvEXO70 and AtEXO70 was one to many, such as the collinear relationships between VvEXO70-08 and AtEXO70H5, AtEXO70H6, AtEXO70H7 and AtEXO70H8 were be in varying degrees.

Figure 6 Collinearity analysis of the EXO70 gene family between grape and Arabidopsis thaliana.

The color of the strips represents the extent of similarity and homology among the genes. Blue indicates the lowest similarity, followed by green, orange and red, indicating the increasing similarity.

In the process of gene evolution, mutations include synonymous mutation and non-synonymous mutation, and 3 important values, named the synonymous mutation frequency (dS) and the non-synonymous mutation frequency (dN) and the ratio of dN/dS. The value of dN/dS plays an important role in gene selection and evolution. When dN/dS >1 is the positive selection, dN/dS = 1 is the neutral selection, 0 <  dN/dS < 1 is purifying selection (Yang & Bielawski, 2000;Yadav et al., 2015). To further explore the evolutionary pattern of this gene family, we found that the values of dN/dS of all 23 duplicated gene pairs were less than 1 between grape and A. thaliana, indicating that the evolution pattern was purifying selection, which played a crucial role in keeping the number of EXO70 in grape and could help to maintain the basic function of this gene (Table 2).

Table 2 Selective pressure analysis of VvEXO70 gene family.

Ribbon color	A pair of genes	S	N	dS	dN	dN/dS	
Red	VvEXO70-02∼AtEXO70A1	466.2	1,447.8	1.4522	0.0966	0.0665	
VvEXO70-05∼AtEXO70A2	463.9	1,429.1	1.6586	0.1357	0.0818	
VvEXO70-12∼AtEXO70G1	444.8	1,451.2	2.1721	0.1488	0.0685	
VvEXO70-13∼AtEXO70D2	449.1	1,362.9	4.5850	0.2343	0.0511	
Orange	VvEXO70-01∼AtEXO70C1	446.5	1,407.5	49.3070	0.3641	0.0074	
VvEXO70-03∼AtEXO70E2	495.2	1,406.8	2.8294	0.4196	0.1483	
VvEXO70-05∼AtEXO70A3	146.4	1,323.6	2.1672	0.3704	0.1709	
VvEXO70-07∼AtEXO70B1	348.6	1,001.4	2.5887	0.1652	0.0638	
VvEXO70-13∼AtEXO70D1	452.6	1,362.4	3.6151	0.2335	0.0646	
VvEXO70-13∼AtEXO70D3	450.1	1,355.9	2.8325	0.2158	0.0762	
Green	VvEXO70-01∼AtEXO70C2	421.2	1,483.8	6.1042	0.3901	0.0639	
VvEXO70-04∼AtEXO70F1	415.3	1,516.7	5.6402	0.2153	0.0382	
VvEXO70-06∼AtEXO70H3	470.6	1,380.4	3.1632	0.4221	0.1334	
VvEXO70-06∼AtEXO70H4	480.9	1,382.1	39.8698	0.3318	0.0083	
VvEXO70-07∼AtEXO70B2	369.1	950.9	3.8067	033113	0.0818	
VvEXO70-08∼AtEXO70H5	446.4	1,323.6	13.2657	0.4503	0.0339	
VvEXO70-08∼AtEXO70H7	463.4	1,354.6	5.5653	0.3917	0.0704	
VvEXO70-10∼AtEXO70G2	489.8	1,448.2	4.1836	0.5181	0.1238	
VvEXO70-11∼AtEXO70H1	450.4	1,343.6	4.0569	0.3224	0.0795	
VvEXO70-11∼AtEXO70H2	468.3	1,331.7	2.6233	0.3326	0.1268	
VvEXO70-14∼AtEXO70E1	481.7	1,438.3	7.4183	0.3943	0.0531	
Blue	VvEXO70-08∼AtEXO70H6	440.3	1,335.7	17.8262	0.4608	0.0258	
VvEXO70-08∼AtEXO70H8	444.5	1,256.5	34.1171	0.4399	0.0129	
Notes.

‘S’ represents the numbers of synonymous, ‘N’ represents non-synonymous, ‘dS’ represents the synonymous mutation frequency and dN represents the non-synonymous mutation frequency and the ratio of dN/dS.

Analysis of base number and codon usage bias of VvEXO70s

RSCU is the ratio of the actual value used by the codon to the theoretical value used by that (Sharp & Li, 1986; Sharp & Li, 1987). The theoretical value is the value with the same frequency of codon, that is, there is no codon bias. If RSCU=1, the use of codon has no preference. If RSCU>1, it indicates that the codon is used more frequently than other synonymous codons, named a high-frequency codon. If RSCU<1, the codon is used less frequently than other codons (Lin et al., 2002; Li et al., 2016). RSCU values results (Table S6) of VvEXO70s codon showed that 14 VvEXO70 genes contained a total of 8871 codons (excluding the termination codon). Among them, there were 5984 codons’ RSCU>1, 2620 codons ending in A/U, and 3364 codons ending in C/G, accounting for 43.78% and 52.22% of the total codons of RSCU>1, respectively. This suggested that the codon ending in C/G was the preferred codon of the EXO70 gene family. Among all codons, the RSCU value of AGG codon of VvEXO70-10 gene was the highest (3.18). Some RSCU values of codons ending in G/C differed from others, for example, RSCU values of UUC codons of VvEXO70-01, VvEXO70-04, VvEXO70-06, VvEXO70-07 and VvEXO70-08 were greater than 1, while RSCU values of other UUC codons were less than 1.

Analysis of the codons usage parameters of VvEXO70s is shown in Table S7. ENC, CAI and CBI values were used to predict gene expression levels. The ENC value of the effective codon ranges from 20 to 61, and the smaller the value, the stronger the applicability of the gene codon (Gupta, Bhattacharyya & Ghosh, 2004; Satapathy et al., 2017). When the use bias reaches the maximum degree, the ENC value is 20; The smaller the usage bias, the closer to 61 (Wright, 1990). The higher the codon preference was, the higher the gene expression level was, and the ENC value was negatively correlated with CAI and CBI values. Low expression gene has low preference, and its CAI and CBI values are small, which will be affected by the amino acid composition and gene length of the gene (Wright, 1990; Marashi & Najafabadi, 2004). Furthermore, we found the average of ENC of VvEXO70s was 54.84, the minimum value was 53.2 (VvEXO70-10), and the maximum value was 59.07 (VvEXO70-07), indicating that the codon preference of this gene family was weak and the usage degree of deviation from random selection was relatively consistent. In general, CAI and CBI values were positively correlated, with values ranging from 0 to 1. Genes, with higher expression, had higher CAI and CBI values. The closer to 1, the higher the value, and the stronger the codon preference (Wu et al., 2007). In addition, the CAI mean of VvEXO70s was 0.203 and the CBI mean was −0.031. The above results indicated that VvEXO70 gene family had weak codon preference. Among them, CAI and CBI values of VvEXO70-01 were the maximum values, which were 0.267 and 0.102 respectively. In addition, the Fop value of VvEXO70-01 gene was the maximum value among all members, indicating that the gene contained the codon with the highest frequency of usage. Further analysis of GC contents of 14 VvEXO70 genes and GC, A, T, C and G in the third place of the codon showed that the average G3s content was 36.97% and the average C3s content was 29.29%. The average content of T3s and A3s was 34.94% and 26.71%, respectively. The average GC content was 46.80%, and the average GC3s content was 50.60%. This indicated that the base selection of the third position of the code of VvEXO70s had a weak C/G preference, and the codon ending in C/G was preferred.

Correlation coefficient between parameters of VvEXO70s codon usability (Table S8) showed that CAI values, CBI values and FOP values were extremely positively correlated with each other (p < 0.01). C3s values were extremely positively correlated with CBI, Fop, GC3s and GC values (p < 0.01). GC3s values were extremely positively correlated with GC values (p < 0.01). L_sym and L_aa values were extremely positively correlated (p < 0.01). GC values were positively correlated with CBI and Fop values (p < 0.05). T3s values were extremely negatively correlated with C3s, CBI, Fop, GC3s and GC values (p < 0.01). A3s values were extremely negatively correlated with C3s and GC3s values (p < 0.01), and negatively correlated with GC values (p < 0.05). ENC values were negatively correlated with L_sym and L_aa values (p < 0.05). It showed that the base content in the third position of the synonymous codon had a high influence on the gene expression level and codon preference (Zhang et al., 2011).

Microarray of VvEXO70s

Expression patterns of all VvEXO70 from transcriptomic data from grape organs and tissues at different stages of development and under abiotic stress were studied (Fig. 7). The data were mainly from a total of 54 grapevine samples, covering most of the grape organs at different stages, were collected. Three biological replicates were taken for each sample, resulting in a total of 162 observations. The collected plant organs were: bud, inflorescence, tendril, leaf, stem, root, developing berry, withering berry, seed, rachis, anther, carpel, petal, pollen, and seedling. The results showed that most genes are expressed to varying degrees in tendrils, leaves, seeds, buds, roots and stems. Among them, VvEXO70-01 was mainly found in pollen and stamens, with the highest expression in pollen. VvEXO70-02 was mainly found in pericarp-veraison, seed-veraison and skin-veraison. VvEXO70-04 was detected in bud-winter dormant, pericarp-veraison, seed-veraison and flesh-PHWI, and its expression level was relatively higher than that of other genes. In all the tissues, VvEXO70-05 had the highest content in stamens, followed by pollen and flower-well developed. The expression of VvEXO70-11 was the highest in the leaf-fruit set. VvEXO70-13 had the highest expression in flesh-PHWIII. However, VvEXO70-04 did not expressed in pollen, bud, leaf, seedling and skin-ripening. VvEXO70-14 did not expressed in skin-rippen, flesh and pericarp. All VvEXO70-07, VvEXO70-08, VvEXO70-09 and VvEXO70-10 didn’t expressed in leaf-sensecent, seedling, bud, carpel, pollen, seed, flesh-ripening, pericarp and skin-PHWI, -PHWII, -PHWIII.

Figure 7 The expression pattern of VvEXO70 gene family in different tissues and organs.

The band indicates the level of gene expression.

Figure 8 Analysis of qRT-PCR expression results of VvEXO70 gene family.

14 VvEXO70 genes were analyzed by qRT-PCR which were used to assess VvEXO70 relative expression levels in leaves sampled at 12 h and 24 h. A∼ N represent VvEXO70 -01∼VvEXO70 -14 gene, respectively. Yellow and blue represent the relative expression for 12 h and 24 h, respectively.

qRT-PCR analysis of VvEXO70s

This study further verified and analyzed the relative expression characteristics of VvEXO70 s. Figure 8 showed that there were significant differences in the relative expression levels of part of genes when all VvEXO70s genes were induced by different hormones and abiotic stresses about 12 h and 24 h. The relative expression levels of all genes in 5 mM SA, 0.1 mM MeJA, 20% PEG6000 and 4 °C for 24 h were all higher than that for 12 h. Compared to the CK, the relative expression quantity of VvEXO70-01, 02, 03, 04, 05, 06, 09, 12, 13, and 14 up-regulated under 2 mM ABA treatment about 12 h and 24 h. On the contrary VvEXO70-07’s relative expression was significantly down-regulated. After 20% PEG6000 treatment for 24 h, the relative expression levels of VvEXO70-02 and VvEXO70-05 significantly up-regulate and were 361 and 219 times higher than that of CK respectively. When the leaves were subjected to 4 °C, VvEXO70-05 had the highest relative expression about 24 h. With 2 mM ABA about 12 h, the relative expression of VvEXO70-05 was highest. VvEXO70-13’s relative expression with 5 mM SA about 24 h was the highest. Most of the genes were up-regulated after 0.1 mM MeJA, especially VvEXO70-05’s at 24 h. Under 7 mM H2O2, all genes’ expression for 12 h were higher than 24 h, among which VvEXO70-02, 03, 04, 05, 06 and 14 significantly up-regulated. This result showed that many VvEXO70 genes were likely to play critical roles in the abiotic and hormonal stress signaling transduction pathways.

Discussion

Exocyst belongs to one of ancient families of the eukaryotic tetraploid system, which is an octameric vesicle mesenteric complex acting upstream of SNARE mediated fusion of extracellular vesicles with the plasma membrane (Whyte & Munro, 2002; Zhang et al., 2010). EXO70, a subunit of the exocyst, together with SEC3 depends on PIP2 in the cell membrane localization. Under the synergistic action of these three proteins, a cell membrane target polarity site is established and then interacted with other 6 exocrine complex components to tether exocrine vesicles to the cell membrane (He et al., 2007; Zhang et al., 2008; Li et al., 2010). This study has found that VvEXO70 gene family consisted of 7 subfamilies (Fig. 4A), which could trace to three ancient EXO70 genes existing in common ancestor of mosses and vascular plants, named EXO70.1, EXO70.2 and EXO70.3. These genes were independently replicated in mosses, lycopodium, and angiosperms lineages (Synek et al., 2006; Chong et al., 2010; Fatima et al., 2012).

In this experiment, a total of 14 VvEXO70 genes were obtained by bioinformatics and domain comparison, which were named VvEXO70-01 ∼VvEXO70-14. Grape, A. thaliana, rice protein sequence were used to construct a phylogenetic tree divided into EXO70A ∼J (Fig. 3), in which EXO70F, H and J did not contain VvEXO70 gene, but only OsEXO70 gene, which was the same as previous research groups (Fatima et al., 2012; Yang et al., 2015a). There were differences in the distribution of exons in the same subgroup, which was similar to the results of Yang et al. (2015b) in cabbage and Chinese cabbage and this suggested that EXO70 genes may have some similarities in the evolution of different species. Multiple sequence alignment (Fig. 4D) and motif analysis (Fig. 4C) showed that VvEXO70 proteins contained a complete pfam03081 domain, which was composed of most of the α-helix and was a key factor in determining the function of EXO70 proteins. Dellago et al. (2011) found that EXO70 could participate in pre-mRNA clipping, and all known EXO70 interacted with other proteins. For this, the conservative pfam03081 may be a necessary domain for interaction with other proteins, but its specific function has not been further studied (Yang et al., 2015a) Besides, the domain had two highly fractured bases, A and B (Fig. 4D). The remaining genes contained all motifs except for VvEXO70-07 and VvEXO70-12, so we predicted that this gene family could have high structural similarity and conservatism in the evolutionary process, and certain variation. It was found that the C-terminal of EXO70 gene was the most conserved structural domain in the whole molecule, which contained a large number of basic residues such as Arg and Lys. It was close to the plasma membrane and further associated with Rho3p through the interaction between the clustered alkaline residues and Rho3p (Dong et al., 2005; Hamburger et al., 2006).

Understanding the protein’s subcellular location information (Table S4) may provide us with the necessary help to infer the biological function of the protein, VvEXO70s woud mainly locate in nucleus, cytoplasm, chloroplast, mitochondria and plasma membrane so it was speculated to be related to plant cytoplasmic secretion, photosynthesis, respiratory action and cell growth and development. Plant tissue expression specificity (Fig. 7) found that most of VvEXO70 gene were expressed to varying degrees in pollen, tendrils, leaves, seeds, buds, roots and stems. Among them, VvEXO70-01 was mainly found in pollen and stamens. AtEXO70A1 expresses in all plant tissues by gene chip and qRT-PCR (Synek et al., 2006; Chong et al., 2010). Moreover, Li et al. (2010) also found that AtEXO70 gene mainly expresses in tissues with high exocrine activity, such as elongated cells at growth points and developed xylem molecules, while the expression was low in other tissues. Yang & Bielawski (2000) detected BoEXO70A1 expression in stem, leaf, petal, anther, stigma, style and ovary by the Northern method, indicating that it may be a constitutive expression gene, but it still needs to be further detected. These indicated that the EXO70 genes family may be related to the tip growth and elongation development of plants.

The collinear relationship (Fig. 6) between grape and A. thaliana EXO70 genes was analyzed, and 23 pairs of collinear genes were found between grape and A. thaliana. Also, the dN/dS values of these gene replication pairs were all less than 1, indicating that they had been purified and selected to predict their high conservatism in the evolution process (Table 2). The analysis of codon bias (Table S7) showed that the codon bias of the members of the gene family was weak and the degree of deviation from random selection was consistent. However, Yang et al. (2015a) and Yang et al. (2015b) found that the codon bias of cabbage and chinese cabbage was almost identical, while that of A. thaliana and rice was significantly different. Besides, according to the correlation of each parameter, the base content in the third position of the synonymous codon has a high influence on the gene expression level and codon preference, it was similar to Zhang et al,. found in 2011. This suggested that the result of using codon preference in the evolutionary process to predict the genomic location of EXO70 for unknown proteins in different species may be different.

When plants are under stresses such as low temperature, drought and hormone, these stress signals will be converted into signal factors, activate the cis-acting elements of transcription factor binding genes and stimulate the expression of related genes and responding to stresses (Shinozaki & Shinozaki, 1997; Teixeira et al., 2004; Hadiarto & Tran, 2011). The 2 kb upstream sequences of VvEXO70s (Fig. 5) found that VvEXO70s contained various cis-acting elements such as G-box and Box-4 elements related to light stress, and MYB and MYC elements related to drought presented in all genes, suggesting that this gene family had certain drought resistance. qRT-PCR (Fig. 8) suggested that VvEXO70s’ relative expression were higher than CK with 20% PEG6000 treatment for 24 h, indicating that most of VvEXO70s had drought resistance, which was similar to the results of cis-acting element. In addition, the VvEXO70s also contained many other different cis-acting elements, such as 28, 30, and 40 components related to SA, MeJA, and ET, respectively. Pieterse et al. (2009) also find that plant hormones SA, MeJA and ET were important regulatory factors in plant resistance signal transduction pathways. It is speculated that these cis-acting elements could have a great role in plant growth and resistance. qRT-PCR (Fig. 8) found that many VvEXO70 genes had different responses to stresses. For example, all genes’ relative expression in 5 mM SA, 0.1 mM MeJA, 20% PEG6000 and 4 °C for 24 h were all higher than that for 12 h. After 20% PEG6000 treatment for 24 h, the relative expression levels of VvEXO70-02 and VvEXO70-05 significantly up-regulate and were 361 and 219 times higher than that of CK respectively. VvEXO70-13’s relative expression under 5 mM SA for 24 h was the highest. Most of genes up-regulatedly expressed with 0.1 mM MeJA, especially VvEXO70-05’s at 24 h. The relative expression level of all genes with 7 mM H2O2 for 12 h were higher than 24 h, among which VvEXO70-02, 03, 04, 05, 06 and 14 significantly up-regulated. VvEXO70-05 had the highest relative expression with 4 °C for 24 h. With 2 mM ABA treatment for 12 h, the relative expression of VvEXO70-05 was the highest and of VvEXO70-07 significantly down-regulate compared to the CK, This showed that many VvEXO70 genes were likely to play critical roles in the abiotic and hormonal stress signaling transduction pathways.

Conclusion

VvEXO70 gene family consists of 14 members, and most genes expressed to varying degrees in tendrils, leaves, seeds, buds, roots and stems. In addition, qRT-PCR analysis showed VvEXO70-02, VvEXO70-03, VvEXO70-04, VvEXO70-05, VvEXO70-13 and VvEXO70-14 may have certain function in resisting abiotic and hormone stresses. This provides ideas for further exploring the function of VvEXO70s and breeding work of grape in the later stage.

Supplemental Information

Supplemental Information 1 The tertiary structure of VvEXO70 gene family

Click here for additional data file.

Supplemental Information 2 Numbers of genes encoding EXO70 in plant genomes

Click here for additional data file.

Supplemental Information 3 qRT-PCR primers for expression on analysis of VvEXO70 gene family

Click here for additional data file.

Supplemental Information 4 The secondary structure of Vv EXO70 protein sequences

Click here for additional data file.

Supplemental Information 5 Subcellular location prediction of Vv EXO 70 gene family

Click here for additional data file.

Supplemental Information 6 c is-acting elements existed in the 2 kb upstream region of VvEXO70 gene family

Click here for additional data file.

Supplemental Information 7 The Codon number and RSCU value of VvEXO70 gene family

Click here for additional data file.

Supplemental Information 8 Codon usage feature of VvEXO70 gene family

Click here for additional data file.

Supplemental Information 9 Codon usage correlation analysis of VvEXO70 gene family

Click here for additional data file.

Supplemental Information 10 Original qRT-PCR data

Click here for additional data file.

Supplemental Information 11 The amino acid sequence of grape, Arabidopsis thaliana, and rice

Click here for additional data file.

Additional Information and Declarations

Competing Interests

Author Contributions

Data Availability

The authors declare there are no competing interests.

Han Wang conceived and designed the experiments, performed the experiments, analyzed the data, prepared figures and/or tables, authored or reviewed drafts of the paper, and approved the final draft.

Zong-Huan Ma performed the experiments, analyzed the data, authored or reviewed drafts of the paper, and approved the final draft.

Juan Mao conceived and designed the experiments, analyzed the data, authored or reviewed drafts of the paper, and approved the final draft.

Bai-Hong Chen conceived and designed the experiments, authored or reviewed drafts of the paper, and approved the final draft.

The following information was supplied regarding data availability:

Original qRT-PCR data and grape, Arabidopsis thaliana, and rice amino acid sequences are available in the Supplementary Files.

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
