# Peer review of "Genome-wide identification and expression analysis of the EXO70 gene family in grape (Vitis vinifera L)"

_PeerJ, doi:10.7717/peerj.11176_

## Round 0.1 · original submission · Major Revisions

Reviewers have commented on the manuscript and found great merit. Authors are encouraged to address the comments raised. In addition, authors are encouraged to create a new subsection at the start of the materials and methods, called overview of study. In it, the authors should provide a snapshot of the experimental program, supported by a schematic flow diagram. This is to help bring the readers into the work. Authors are encouraged to further justify this study, at the later part of the introduction, before the objective statement.
A brilliant study, look forward to the revised manuscript :)

Reviewer 1 ·

Basic reporting

The abstract describes research context, briefly introduces the research method and states the research outcome and future expectation. In terms of the background context, it generally states the definition of EXO70 gene family, its function and the current inefficiency of knowledge on EXO70 gene family. After that, the abstract describes the process of the experiment and finally puts out the findings of the study comprehensively and coherently. In the final part, it provides us with the purpose and goal of this paper as well which makes sense to us what the authors are going to do.
The introduction generally gives us a detailed description of EXO70 gene family including the definition, functions and its working essence by providing some examples, interpreting its discovery and development history, which I think makes us a relatively complete understanding and knowledge of EXO70 gene family
This part first of all directly tells us two methods used to identify EXO70 genes and then a description of detailed procedures of how they carry out the experiment and do analysis. These processes are aligned in order of experimental logic and each analysis subsection is stated briefly and coherently. Analysis tools and techniques are also displayed in each analysis part so that we could get more idea of how they do it. Methods used are sound and data interpretation is up to the mark.
The results part displays the outcome of the series of modelling analysis and test in the logic order in Methods part. The analysis results are put out by listing concrete data, including numeric data, statistical graphs and figures, in a specific and complete way to me.
The discussion part discusses the findings discovered in the paper, including explaining how the findings come out and interpreting the results in more details, which gives us a supplementary understanding of the results obtained in the paper.
Overall, the paper gives us a detailed understanding of the characteristics of EXO70 gene family by carrying out a series of experiment and statistical analysis in a logical order. Except for some minor errors, the paper is a complete, detailed one that interprets EXO70 gene family and leaves us open research areas to further explore the characteristics of EXO70 genes.

Experimental design

no comment

Validity of the findings

no comment

Additional comments

I have raised some related question which is mentioned in the detailed comments.
Point 1: Authors should carefully recheck the manuscript for the typos and scientific styles. For example, line-17 collinearity and selective pressure, codon bias, tissue expression and the like.
Point 2: The English was mostly understandable.
Point 3: Line 34-35, Please rephrase the sentence.
Line 35, When plants are under stresses, user stress??
Point 4: line 84, “and so on. And through”.?? rephrase
Point 5: In the material and method section, rephrase the heading 2.2
Point 6: line number 136, formatting error.
Point 7: In many paragraph repetitions of these words “In Addition, these results" etc.
Point 8: line 452-459, need to add more explanation with fold change.

·

Basic reporting

The article is well written, although some typos and in few cases a clear form should be used, as mentioned in the comments.
The literature is enough rich
The article is well structurated
The article is an unique body and the studies conducted are coherently with the aims.

Experimental design

The experimental design, the methofs and the tools used are coherent

Validity of the findings

All data are strongly validated

Additional comments

Author you should consider that all the periods from line 195 to 207 are descriptions which deals with putative location of each gene; therefore, You must use a conditional form in the sentences in which you describe the results.
Line 198, delete the point from “…. abundant.in cytoplasm… “
Lines 199-201. Change the phrase from “VvEXO70-02, VvEXO70-03, VvEXO70-05, VvEXO70-07, VvEXO70-10 and VvEXO70-11 didn't exist in chloroplast but VvEXO70-08 was the highest abundant,” TO “VvEXO70-08 scored the highest probability to be present only in the chloroplast”
Line 201, delete the comma after the word abundant and insert the full stop.
Line 206, delete the full stop after the VvEXO70-05
Line 230 delete the word gene
Line 275 and 276, Authors use correct diction of “the synonymous mutation frequency (dN) and the non-synonymous (dS)” therefore change as follow of “the synonymous mutation frequency (ds) and the non-synonymous (dN)”. Correct in all other part of the text.
Line 276 mutation frequency (dS) and the ratio of dN/dS. The value of dN/dS plays an important role in gene
Caption of Figure 5. Correct this sentence in figure’s caption since it is not clear “Red indicates the lowest similarity, followed by orange, green and blue, indicating the decreasing similarity.” To avoid misunderstanding therefore change in “mutation, mainly including the synonymous mutation frequency (dS) and the non-synonymous (dN)”
Line 276 mutation frequency (dS) and the ratio of dN/dS. The value of dN/dS plays an important role in gene
Line 359 you should write in the extended form of the word Fig. 7: figure 7
Line 378: change from substances to proteins
Lines from 385 – 387: adjust better the phrase it is not clear!
Line 3091: what means etc. do you know publication related to other plant species or not?
Along the text after the first mention of Arabidopsis thaliana for the following time use A. thaliana in the italic form.
Caption of figure 7: Author you should insert the statistical analysis and indicate if the replication are generated from three biological replication or not.
Use italic form for all gene name
After the first mention of Arabidopsis thaliana all the time you mention this species you should use the contact form like A. thaliana, written in the italic form

Reviewer 3 ·

Basic reporting

The English language is clear enough but some sections of the article such as the discussion need to be improved.
Introduction and background showing perfectly the context of the study. The literature should be updated since most of them are before 2010, I think is also better to cite more the international published papers in this study.
The structure conforms the PeerJ standard.
Figures are relevant but the quality should be improved.

Experimental design

The research is so interesting and is within the scope of this journal.
Research is well defined and the methods is defined with suficient details.

Validity of the findings

The data provided are statistically sound and are robust. The discussion should be improved in the actual version is more reporting the results. Conclusions are well stated.

---

## Round 0.2 · Major Revisions

Thank you authors for revising your manuscript. After a quick look, it appears the revision was rushed and not carefully prepared.

-Authors are encouraged to create a new subsection at the start of the materials and methods, called overview of study. This should be 2.1. Yes, you provided an overview section, but it should be under Materials and Methods.

-In it, the authors should provide a snapshot of the experimental program, supported by a schematic flow diagram. There was no schematic flow diagram in your revised submission. The overview of study must be supported by a schematic diagram showing the flow of the entire experimental program.

- Authors are encouraged to further justify this study, at the later part of the introduction, before the objective statement. This appears not to have been fully addressed.

Authors should address these, and incorporate them, not only in the manuscript but in the rebuttal, in addition to all the responses to all the reviewers' comments. That is, full details of reply to all reviewers' comments must be in your rebuttal letter in addition to responding to editor comments.

Thank you.

---

## Round 0.3 · Minor Revisions

Thank you authors for revising your work, reviewers have found it worthy of publication.
Editor encourages authors to please attend to the following:
a) At the beginning of paragraph 4 of introduction, kindly add three/ four sentences about the crux of genome-wide identification and expression analysis, and what it entails, specific to the context of this work (Remember, readers of this work, may not necessarily be experts in this field, there could also be learners).

b) Lines 78-83, indicate 'EXO70 genes' after the citation of (Zhao et al., 2018) in Line 83 (Move 'EXO70 genes' mentioned in line 80, to be before the full stop in line 83)

c) Overview of study, diagram is very unique. Please, have two three sentences describing this diagram, and how it connects to the objective of this work. Guide the reader into this work

d) Line 182 - Results (Not 'Results and Analysis'

e) In the discussion section, kindly indicate (Refer to Figure ??) in all the places where one or more figures is being referred to. This is very important, as it will guide readers well.

Look forward to receiving your revised manuscript.
Thank you :)

·

Basic reporting

Now is ready for pubblication

Experimental design

it is ok

Validity of the findings

the area could be the plant molecular physiology

Additional comments

The manuscript was improved

---

## Round 0.4 · accepted · Accept

The editor is very satisfied with all revisions made and considers the revised manuscript now acceptable for publication. The editor believes the authors have greatly benefitted from the review process, which has improved the quality of their work. Thank you authors for your very fine contribution, and for finding PeerJ as your journal of choice. Congratulations and look forward to your future scholarly contributions.